# LoRATv2: Enabling Low-Cost Temporal Modeling in One-Stream Trackers

**Liting Lin**
Pengcheng Laboratory
`lt.lin@outlook.com`

**Heng Fan**
University of North Texas
`heng.fan@unt.edu`

**Zhipeng Zhang**
Shanghai Jiao Tong University
Anyverse Intelligence
`zhipeng.zhang.cv@outlook.com`

**Yuqing Huang**
Pengcheng Laboratory
`domaingreen2@gmail.com`

**Yaowei Wang**[*]
Harbin Institute of Technology, Shenzhen
Pengcheng Laboratory
`wangyaowei@hit.edu.cn`

**Yong Xu**
South China University of Technology
`yxu@scut.edu.cn`

**Haibin Ling**[*]
Westlake University
`linghaibin@westlake.edu.cn`

## Abstract

Transformer-based algorithms, such as LoRAT, have significantly enhanced object-tracking performance. However, these approaches rely on a standard attention mechanism, which incurs quadratic token complexity, making real-time inference computationally expensive. In this paper, we introduce LoRATv2, a novel tracking framework that addresses these limitations with three main contributions. First, LoRATv2 integrates frame-wise causal attention, which ensures full self-attention within each frame while enabling causal dependencies across frames, significantly reducing computational overhead. Moreover, key-value (KV) caching is employed to efficiently reuse past embeddings for further speedup. Second, building on LoRAT's parameter-efficient fine-tuning, we propose Stream-Specific LoRA Adapters (SSLA). As frame-wise causal attention introduces asymmetry in how streams access temporal information, SSLA assigns dedicated LoRA modules to the template and each search stream, with the main ViT backbone remaining frozen. This allows specialized adaptation for each stream's role in temporal tracking. Third, we introduce a two-phase progressive training strategy, which first trains a single-search-frame tracker and then gradually extends it to multi-search-frame inputs by introducing additional LoRA modules. This curriculum-based learning paradigm improves long-term tracking while maintaining training efficiency. In extensive experiments on multiple benchmarks, LoRATv2 achieves state-of-the-art performance, substantially improved efficiency, and a superior performance-to-FLOPs ratio over state-of-the-art trackers. The code is available at `https://github.com/LitingLin/LoRATv2`.

## 1 Introduction

Visual object tracking is a fundamental task in computer vision with broad applications in surveillance, autonomous driving, robotics, augmented reality, and human-computer interaction. Accurate and continuous object localization across frames is crucial for the reliability of these applications.

---

[*]Corresponding author

39th Conference on Neural Information Processing Systems (NeurIPS 2025).

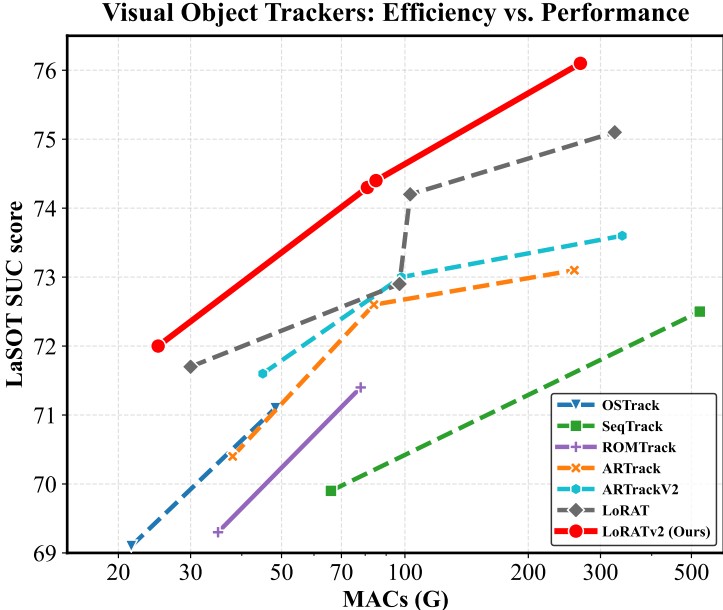

Figure 1: Comparison of our tracker (LoRATv2) with other Transformer trackers on computational complexity (MACs) and performance (SUC scores) on the LaSOT dataset.

Recent advances in Vision Transformers (ViT) [14] have substantially improved tracking performance, primarily due to the powerful representation capabilities of self-attention. In particular, *one-stream* trackers, such as MixFormer[9], OSTrack [43], ARTrack [35], SeqTrack [7], and LoRAT [24], unify feature extraction and relation modeling within a single Transformer backbone. Although this design has demonstrated excellent accuracy and efficiency, these methods rely on *standard* self-attention, which has a quadratic complexity with respect to the number of tokens. Consequently, inference can become prohibitively expensive when dealing with high-resolution input frames or extended temporal sequences, constraining real-time deployment on resource-limited systems.

LoRAT [24], a recently proposed state-of-the-art tracker, employs Parameter-Efficient Fine-Tuning (PEFT) via Low-Rank Adaptation (LoRA) [19] to significantly reduce the *training* overhead for large-scale Transformer trackers. However, it retains the standard self-attention mechanism at *inference* time, leading to substantial computational costs as input resolution or frame count increases.

Furthermore, while one-stream trackers [43, 24] excel with single template-search pairs, simply extending them to multi-frame inputs using standard bidirectional attention often yields only marginal performance gains in our preliminary experiments. We hypothesize this may be partly due to potential rank collapse issues in deep Transformers with bidirectional attention, which can limit their expressive power for complex temporal sequences [13].

To overcome these limitations and better model the sequential nature of tracking, we propose **LoRATv2**. Visual object tracking can naturally be framed as an autoregressive sequence prediction task, where the target's state in the current frame depends on its history. This perspective aligns well with causal attention mechanisms. Therefore, LoRATv2 introduces the *frame-wise causal attention* mechanism within the ViT backbone. In this design, tokens within the current frame maintain full self-attention for rich intra-frame representation, while attending *causally* only to tokens from previous frames, enforcing an autoregressive structure. This approach, coupled with *key-value (KV) caching* [33, 11, 31] to reuse past embeddings, not only significantly enhances inference efficiency by reducing MACs and improving FPS, but as our experiments demonstrate (shown in Fig. 1), also leads to improved tracking performance by effectively modeling temporal dependencies.

The adoption of frame-wise causal attention, while beneficial for temporal modeling and efficiency, introduces an inherent asymmetry in how different input streams (e.g., the static template, the first search frame, subsequent search frames) process information. Unlike traditional Siamese trackers or one-stream trackers with full attention where all tokens potentially interact symmetrically, in our

causal setup each stream has a distinct view of historical context, as it can only attend to itself and preceding information. To manage this asymmetry effectively, while preserving LoRAT's parameter-efficient fine-tuning paradigm, we propose *Stream-Specific LoRA Adapters (SSLA)*. SSLA allocates dedicated LoRA [19] adapters to each input stream, and the main ViT backbone is kept frozen and shared. This allows each stream to develop specialized adaptations. For instance, the template's LoRA can focus on robust initial feature extraction, the first search frame's LoRA might learn to enhance target features while identifying distractors, and LoRAs for subsequent search frames can specialize in precise localization based on the evolving temporal context. This modification maintains the benefits of a shared backbone yet allows minimal, stream-dependent adaptation critical for effective multi-frame tracking.

Finally, we present a *two-phase progressive training* strategy to incrementally expand from single-search-frame to multi-search-frame tracking. This approach significantly reduces the computational resources required for training, as at any given time, only the LoRA modules associated with one or two input streams are trainable, while previously trained LoRA modules and the main backbone remain frozen. This easy-to-hard, curriculum learning-style [2] training paradigm, as demonstrated in our experiments, leads to better tracking performance compared to training all LoRA modules from scratch. Furthermore, this strategy naturally yields a family of trackers capable of processing different numbers of input frames (e.g., a single-search-frame tracker from the initial stage and a multi-search-frame tracker from the extension stage), providing additional flexibility for deployment scenarios with varying computational budgets or accuracy requirements.

Our contributions can be summarized as follows:

1. **Frame-wise Causal Attention:** We integrate frame-wise causal attention into one-stream visual tracking, where tokens attend fully within their frame and causally to past frames. Combined with KV caching to reuse past embeddings and prevent re-encoding, this mechanism *enables computationally efficient long-term temporal modeling for trackers* and improves tracking accuracy by robustly capturing temporal context.

2. **Stream-Specific LoRA Adapters (SSLA):** To handle the asymmetry introduced by causal connections, we equip each input stream (template or search) with its own low-rank LoRA modules. This preserves a shared backbone for all streams while allowing minimal, stream-specific adaptation for improved tracking accuracy, with zero additional inference cost.

3. **Two-Phase Progressive Training:** We first train a single-frame tracker (template → one search frame) and then *incrementally* extend it to multi-frame inputs by adding new LoRA adapters only for the additional search frames. This curriculum-like approach improves long-term performance while greatly reducing memory and training time compared to direct multi-frame training from scratch.

4. **Extensive experimental evaluations** across multiple benchmark datasets demonstrate that our proposed approach delivers superior tracking performance and significantly improves performance-to-FLOPs ratio compared to existing state-of-the-art self-attention-based methods.

## 2 Related Work

**Temporal modeling in tracking.** Temporal modeling in visual tracking typically involves leveraging historical information and explicit temporal dependencies to enhance robustness and accuracy. Some methods employ historical prompts to improve tracking stability, such as AQATrack [40] and HipTrack [3]. Approaches like STARK [41] and TATrack [18] dynamically update tracking templates based on prior tracking outcomes to maintain effectiveness over time. Furthermore, autoregressive prediction strategies have been explored extensively in methods such as ARTrack [35], ARTrackV2 [1], and SeqTrack [7], which explicitly model temporal dependencies across frames. Recent advancements also include multi-frame modeling methods. For example, TCTrack [5] incorporates video-level contextual information through adaptive convolutional techniques, whereas VideoTrack [37] uses transformer-based architectures to integrate broader temporal context. Additionally, ODTrack [45] explicitly propagates token sequences between frames to maintain dense contextual associations.

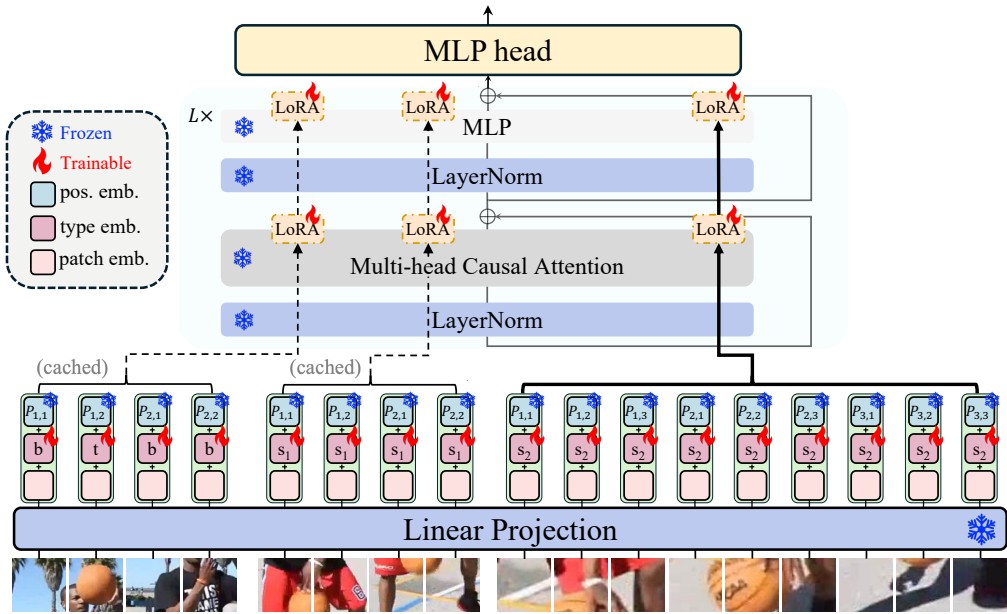

Figure 2: **Overall architecture of LoRATv2.** We tokenize the template and search regions, append them into a single token sequence with shared positional and stream-specific token type embeddings, and feed them to a frozen ViT encoder. Frame-wise causal attention ensures that each frame only attends to itself and preceding frames. Meanwhile, KV caching reuses previously computed key/value embeddings to avoid redundant computation across frames. We attach lightweight LoRA modules (one per stream) to each linear projection layer and keep the main ViT weights frozen. The final embeddings are passed through an MLP head for target classification and bounding-box regression, yielding an efficient and scalable solution for multi-frame visual tracking.

**Causal attention in vision.** Causal attention in vision addresses spurious correlations by explicitly modeling causal relationships. Methods like CATT [42] introduce causal interventions within attention modules to mitigate contextual bias, enforcing object-context separation through sample-wise masking. Building upon these foundations, subsequent approaches extend causal attention to autoregressive frameworks. Vision-RWKV [15], for example, adapts causal attention mechanisms for bidirectional global interactions within vision transformers. To enhance scalability and efficiency, some models incorporate hierarchical or sequential structures. VAR [32] employs causal masking strategies for efficient coarse-to-fine image generation. Similarly, Causal Vision Transformers [22] apply sequential causal attention to efficiently handle large-scale images. Moreover, Show-O [39] unifies multimodal generation tasks under a single causal transformer framework. Collectively, these advancements underscore the versatility and efficacy of causal attention methods across diverse visual tasks, spanning recognition to generation, while enabling scalable and controllable visual modeling.

## 3 Method

Our proposed **LoRATv2** builds upon *LoRAT* [24] and introduces additional components for efficient multi-frame, temporal-aware visual object tracking. First, we revisit LoRAT to recap its core design principles (Sec. 3.1). Next, we present our *Frame-Wise Causal Attention* (Sec. 3.2) and *Key-Value Caching* (Sec. 3.3), which jointly enable efficient multi-frame modeling without re-encoding past frames. To address the *asymmetric* dependency introduced by causal attention, we propose *Stream-Specific LoRA Adapters* (Sec. 3.4), which preserve a unified embedding space while enabling minimal, per-stream adaptation. Finally, Sec. 3.5 details our *Two-Phase Progressive Training* pipeline for scaling from single-frame to multi-frame scenarios with minimal overhead. Unless otherwise noted, all other design and training settings (e.g., shared positional embeddings, token-type embeddings, anchor-free MLP heads) follow LoRAT [24].

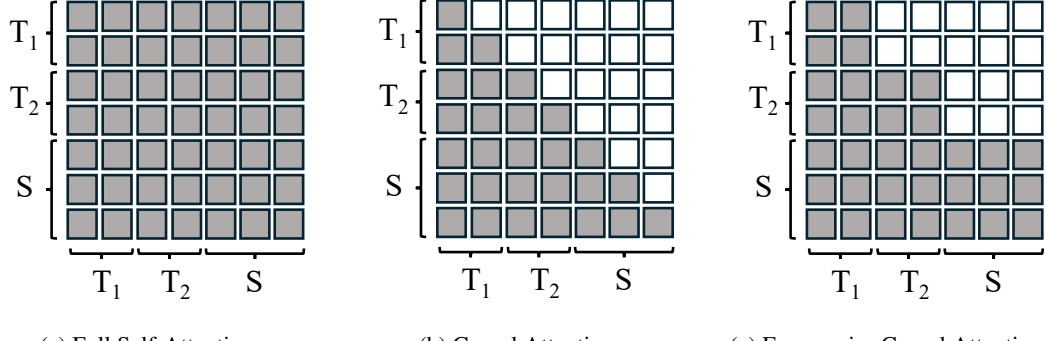

|  (a) Full Self-Attention | (b) Causal Attention | (c) Frame-wise Causal Attention |

Figure 3: Visualization of three attention masks. **(a)** Full self-attention, where each query position can attend to all key positions. **(b)** Causal attention, where each token can only attend to itself and tokens to its left. **(c)** Frame-wise causal attention, where each token can only attend to the tokens within current frame and tokens in the past frames. Dark squares indicate valid (unmasked) attention, while white squares indicate invalid (masked) attention.

## 3.1 Revisiting LoRAT

**One-Stream Transformer Tracker.** LoRAT [24] follows the *one-stream* Transformer tracker paradigm, where the template and the search region are processed together by a single ViT backbone. First, the images are divided into non-overlapping patches. These patches are then flattened and mapped to patch embeddings via a linear projection, yielding the template tokens $\{\mathbf{z}_i\}_{i=1}^m$ and search region tokens $\{\mathbf{x}_j\}_{j=1}^n$. To distinguish these tokens after concatenation, LoRAT incorporates learnable *token type embeddings* that are added to the patch embeddings of each stream (template or search). Additionally, it uses shared positional embeddings for both streams. The final input sequence is formed by concatenating the enriched tokens:

$$\mathbf{X} = [\,\mathbf{z}_1, \ldots, \mathbf{z}_m, \ \mathbf{x}_1, \ldots, \mathbf{x}_n\,]. \tag{1}$$

A Transformer encoder then applies multi-head self-attention and feed-forward layers over $\mathbf{X}$, capturing intra-frame and inter-frame relationships. Finally, the output embeddings corresponding to the search region are fed into an MLP-based head network for target classification and bounding box regression, producing the tracker's output.

**Self-Attention Complexity.** LoRAT employs standard *full* self-attention in its ViT backbone. Let $\mathbf{X} \in \mathbb{R}^{(m+n)\times d}$ denote the token embeddings. The standard attention mechanism first computes:

$$\mathbf{Q} = \mathbf{X}\mathbf{W}^Q, \quad \mathbf{K} = \mathbf{X}\mathbf{W}^K, \quad \mathbf{V} = \mathbf{X}\mathbf{W}^V, \tag{2}$$

and then:

$$\text{Attention}(\mathbf{Q}, \mathbf{K}, \mathbf{V}) \ = \ \text{softmax}\!\left(\frac{\mathbf{Q}\mathbf{K}^\top}{\sqrt{d}}\right)\mathbf{V}. \tag{3}$$

This incurs $\mathcal{O}((m+n)^2)$ operations. Although LoRAT uses Low-Rank Adaptation (LoRA) [19] to reduce *training* overhead, the *inference* cost remains governed by full self-attention.

## 3.2 Frame-Wise Causal Attention

Standard full self-attention (Eq. 3) grows quadratically with the number of tokens and does not incorporate temporal order. To efficiently model longer sequences in tracking, we introduce a causal mask that enforces an autoregressive dependency across frames while preserving full attention within each frame.

Assume we have $T$ frames, each producing $n_t$ tokens $\{\mathbf{x}_1^t, \ldots, \mathbf{x}_{n_t}^t\}$ for $t = 1, \ldots, T$. Concatenate them into a single sequence:

$$\mathbf{X} = \Big[\underbrace{\mathbf{x}_1^1, \ldots, \mathbf{x}_{n_1}^1}_{\text{frame 1}}, \ \underbrace{\mathbf{x}_1^2, \ldots, \mathbf{x}_{n_2}^2}_{\text{frame 2}}, \ \ldots, \ \underbrace{\mathbf{x}_1^T, \ldots, \mathbf{x}_{n_T}^T}_{\text{frame } T}\Big]. \tag{4}$$

In *frame-wise causal attention*, queries from frame $t$ can attend to tokens in frames $1, \ldots, t$ but not to frames $> t$. We implement this via a causal mask $\mathbf{M} \in \mathbb{R}^{N \times N}$, where $N = \sum_{t=1}^{T} n_t$, defined by

$$\mathbf{M}_{(p,q)} = \begin{cases} 0, & \text{if frame}(q) \leq \text{frame}(p), \\ -\infty, & \text{otherwise.} \end{cases} \tag{5}$$

Let $\mathbf{Q} = \mathbf{X}\mathbf{W}^Q$, $\mathbf{K} = \mathbf{X}\mathbf{W}^K$, and $\mathbf{V} = \mathbf{X}\mathbf{W}^V$ be the usual linear projections. Our **frame-wise causal attention** modifies Eq. 3 by adding $\mathbf{M}$ inside the softmax:

$$\text{Attention}(\mathbf{Q}, \mathbf{K}, \mathbf{V}) = \text{softmax}\left(\frac{\mathbf{Q}\mathbf{K}^\top}{\sqrt{d}} + \mathbf{M}\right)\mathbf{V}. \tag{6}$$

Within each frame, $\mathbf{M}_{(p,q)} = 0$ since frame$(p) = $ frame$(q)$, thus preserving *full* (unmasked) attention. Across frames, any future-to-past connection is disallowed by setting $\mathbf{M}_{(p,q)} = -\infty$ if frame$(p) < $ frame$(q)$, enforcing auto-regressive structure. The visualization of frame-wise causal attention mask is shown in Fig. 3c.

### 3.3 Key-Value Caching

When performing online tracking or processing long sequences, naively recomputing key/value embeddings for past frames is computationally expensive. *Key-Value (KV) caching* [33, 11, 31] provides a simple yet powerful mechanism to amortize this cost over time.

At frame $t$, we first compute queries for the current frame:

$$\mathbf{Q}^t = \mathbf{X}^t\mathbf{W}^Q. \tag{7}$$

Here, $\mathbf{X}^t \in \mathbb{R}^{n_t \times d}$ denotes the tokens of frame $t$. The key/value pairs for frame $t$ are

$$\mathbf{K}^t = \mathbf{X}^t\mathbf{W}^K, \quad \mathbf{V}^t = \mathbf{X}^t\mathbf{W}^V. \tag{8}$$

We then *cache* all past keys and values (frames 1 through $t-1$):

$$\overline{\mathbf{K}}^{1:t-1} = \left[\mathbf{K}^1, \ldots, \mathbf{K}^{t-1}\right], \quad \overline{\mathbf{V}}^{1:t-1} = \left[\mathbf{V}^1, \ldots, \mathbf{V}^{t-1}\right]. \tag{9}$$

For the **frame-wise causal attention** at step $t$, we form

$$\widetilde{\mathbf{K}}^t = \left[\overline{\mathbf{K}}^{1:t-1}, \mathbf{K}^t\right], \quad \widetilde{\mathbf{V}}^t = \left[\overline{\mathbf{V}}^{1:t-1}, \mathbf{V}^t\right]. \tag{10}$$

Hence, the attention computation at frame $t$ becomes

$$\text{Attention}\left(\mathbf{Q}^t, \widetilde{\mathbf{K}}^t, \widetilde{\mathbf{V}}^t\right), \tag{11}$$

where the mask $\mathbf{M}$ (Eq. 5) ensures that $\mathbf{Q}^t$ can only attend to $\overline{\mathbf{K}}^{1:t-1}$ (past frames) and $\mathbf{K}^t$ (the current frame), but not beyond. Crucially, we never re-encode past frames; we simply reuse their cached $\mathbf{K}^t, \mathbf{V}^t$.

### 3.4 Stream-Specific LoRA Adapters (SSLA)

While frame-wise causal attention preserves within-frame symmetry, the Siamese template and search region streams *across frames* might still diverge in how they attend to historical context. Specifically, the template in frame $t$ remains mostly static (or slowly updated if multiple templates are used), whereas the search region tokens in frame $t$ accumulate knowledge from frames $\{1, \ldots, t-1\}$. This discrepancy can degrade matching accuracy if not handled carefully.

We therefore introduce *Stream-Specific LoRA Adapters*. Each input stream $s$—template, search$_1$, search$_2$, and so forth—is assigned its own LoRA offset:

$$s \in \{\text{template}, \text{search}_1, \text{search}_2, \ldots\}. \tag{12}$$

Concretely, for the query projection,

$$\mathbf{Q}_s = \mathbf{X}_s\left(\mathbf{W}^Q + \Delta\mathbf{W}_s^Q\right), \tag{13}$$

where $\mathbf{X}_s$ is the set of tokens from stream $s$, $\mathbf{W}^Q$ is the frozen backbone weight, and $\Delta\mathbf{W}_s^Q$ is the learnable low-rank offset for that stream. Analogous offsets $\Delta\mathbf{W}_s^K, \Delta\mathbf{W}_s^V$ attach to $\mathbf{W}^K, \mathbf{W}^V$.

Such design brings the following benefits:

- It preserves a shared representation space via the unmodified backbone.
- It applies minimal, stream-dependent adaptations for each set of tokens.
- It allows easy extension by adding new LoRA modules for additional frames.

Table 1: Benchmarking our tracker on four large-scale challenging datasets. For GOT-10k evaluation, all the methods follow the one-shot protocol, training only on the train split of GOT-10k. **Bold** indicates the best results and underline indicates the second-best.

| Tracker | LaSOT [16] | | | TNL2K [34] | | GOT-10k [20] | | | VastTrack [30] | |
|---|---|---|---|---|---|---|---|---|---|---|
| | SUC | $P_{Norm}$ | P | SUC | P | AO | $SR_{0.5}$ | $SR_{0.75}$ | SUC | P |
| TransT$_{256}$ [8] | 64.9 | 73.8 | 69.0 | 50.7 | 51.7 | 67.1 | 76.8 | 60.9 | 29.9 | 25.4 |
| AutoMatch$_{255}$ [44] | 58.3 | - | 59.9 | 47.2 | 43.5 | 65.2 | 76.6 | 54.3 | 28.8 | 26.6 |
| STARK$_{320}$ [41] | 67.1 | 77.0 | - | - | - | 68.8 | 78.1 | 64.1 | 33.4 | 30.8 |
| KeepTrack$_{480}$ [27] | 67.1 | 77.2 | 70.2 | - | - | - | - | - | - | - |
| MixFormer$_{384}$ [9] | 70.1 | 79.9 | 76.3 | - | - | - | - | - | - | - |
| SBT$_{224}$ [38] | 66.7 | - | 71.1 | - | - | 70.4 | 80.8 | 64.7 | - | - |
| AiATrack$_{320}$ [17] | 69.0 | 79.4 | 73.8 | - | - | 69.6 | 80.0 | 63.2 | - | - |
| SimTrack$_{384}$ [6] | 70.5 | 79.7 | - | 55.6 | 55.7 | 69.8 | 78.8 | 66.0 | 34.4 | 30.3 |
| OSTrack$_{384}$ [43] | 71.1 | 81.1 | 77.6 | 55.9 | 56.7 | 73.7 | 83.2 | 70.8 | 33.6 | 31.5 |
| SwinTrack$_{384}$ [25] | 71.3 | - | 76.5 | 55.9 | 57.1 | 72.4 | 80.5 | 67.8 | 33.0 | 30.3 |
| DropTrack$_{256}$ [36] | 71.8 | 81.8 | 78.1 | 56.9 | 57.9 | 75.9 | 86.8 | 72.0 | 37.0 | 36.5 |
| SeqTrack-B$_{256}$ [7] | 69.9 | 79.7 | 76.3 | 56.4 | - | 74.7 | 84.7 | 71.8 | - | - |
| SeqTrack-L$_{384}$ [7] | 72.5 | 81.5 | 79.3 | 57.8 | - | 74.8 | 81.9 | 72.2 | 39.6 | 40.2 |
| ARTrack-B$_{256}$ [35] | 70.4 | 79.5 | 76.6 | 59.8 | - | 73.5 | 82.2 | 70.9 | - | - |
| ARTrack-L$_{384}$ [35] | 73.1 | 82.2 | 80.3 | 60.3 | - | 78.5 | 87.4 | 77.8 | 35.6 | 32.4 |
| ARTrackV2-B$_{256}$ [35] | 71.6 | 80.2 | 77.2 | 59.2 | - | 75.9 | 85.4 | 72.7 | - | - |
| ARTrackV2-L$_{384}$ [35] | 73.6 | 82.8 | 81.1 | 61.6 | - | **79.5** | **87.8** | **79.6** | - | - |
| CiteTracker$_{384}$ [23] | 69.7 | 78.6 | 75.7 | 57.7 | 59.6 | 74.7 | 84.3 | 73.0 | - | - |
| ROMTrack$_{384}$ [4] | 71.4 | 81.4 | 78.2 | - | - | 74.2 | 84.3 | 72.4 | 37.0 | 36.1 |
| MixViT-L$_{384}$ [10] | 72.4 | 82.2 | 80.1 | - | - | 75.7 | 85.3 | 75.1 | 39.5 | 39.8 |
| ODTrack-L$_{384}$ [45] | 74.0 | 84.2 | 82.3 | 61.7 | - | 78.2 | 87.2 | 77.3 | - | - |
| LoRAT-B$_{224}$ [24] | 71.7 | 80.9 | 77.3 | 58.8 | 61.3 | 72.1 | 81.8 | 70.7 | 38.7 | 37.8 |
| LoRAT-L$_{378}$ [24] | 75.1 | 84.1 | 82.0 | 62.3 | 67.0 | 77.5 | 86.2 | 78.1 | 43.9 | 45.8 |
| LoRATv2-B$_{224}$ | 72.0 | 81.3 | 77.9 | 59.6 | 62.7 | 74.7 | 84.7 | 72.7 | 39.1 | 38.7 |
| LoRATv2-B$_{378}$ | 74.3 | 83.6 | 80.9 | 60.9 | 64.9 | 75.8 | 85.7 | 75.4 | 40.6 | 40.8 |
| LoRATv2-L$_{224}$ | 74.4 | 83.8 | 81.2 | 61.8 | 66.7 | 76.9 | 86.3 | 76.4 | 42.0 | 43.3 |
| LoRATv2-L$_{378}$ | **76.1** | **85.1** | **83.1** | **62.4** | **67.7** | 78.2 | 86.8 | 79.1 | **44.2** | **46.7** |

## 3.5 Two-Phase Progressive Training

Inspired by curriculum learning [2], we adopt a *two-phase* approach to gradually introduce multi-frame complexity. This strategy not only reduces memory requirements compared to training on multiple frames from scratch but also consistently yields better final accuracy in practice.

**Phase 1: Single Template + Single Search Region.** We follow LoRAT [24] by training only two LoRA adapters, one for the template and one for a single search region, while keeping the ViT backbone frozen. This phase converges quickly and requires minimal memory.

**Phase 2: Extending to an Additional Search Region.** Next, we fix all previously learned parameters and attach a new LoRA adapter for the second search region. Only this newly added adapter is updated, enabling efficient multi-frame modeling without large-scale retraining. The final model supports both single-frame (for speed) and multi-frame inference (for improved robustness), with additional frames accommodated via further adapters as needed.

## 4 Experiments

### 4.1 Implementation Details

All models are trained on 4×NVIDIA GeForce RTX 4090 GPUs and evaluated on an NVIDIA GeForce RTX 5090 GPU. Basically, we follow LoRAT [24] for fundamental settings (*e.g.* training datasets, optimization hyperparameters).

**Model Variants.** We develop four LoRATv2 variants using ViT-Base (B) and ViT-Large (L) backbones, trained progressively as described in Sec. 3.5. The specific configurations are:

- **LoRATv2-B/L-224 (Phase 1 Models):**
  - Backbone: ViT-Base/ViT-Large
  - Template ($\mathbf{z}$): $224 \times 224$
  - Search Region ($\mathbf{x}^1$): $224 \times 224$
- **LoRATv2-B/L-378 (Phase 2 Models):**
  - Backbone: ViT-Base/ViT-Large
  - Template ($\mathbf{z}$): $224 \times 224$
  - Past Search Region ($\mathbf{x}^1$): $224 \times 224$
  - Current Search Region ($\mathbf{x}^2$): $378 \times 378$

**Training.** We use LaSOT [16], TrackingNet [28], GOT-10k [20] (excluding 1k sequences as in [21]), and COCO [26] for training. For the GOT-10k evaluation, models are trained exclusively on the GOT-10k training split.

*Phase 1 ($-224$ variants)*: Models are trained for 170 epochs (131,072 iterations/epoch) on ($\mathbf{z}, \mathbf{x}^1$) pairs. The template ($\mathbf{z}$) and search region $\mathbf{x}^1$ are sampled from the same video (up to a 100-frame gap), with strong crop jitter applied to $\mathbf{x}^1$. The ViT backbone (DINOv2 pre-trained [29, 12]) is frozen; two sets of LoRA modules (rank $r = 64$), one for the template stream and one for the search region stream, are trained.

*Phase 2 ($-378$ variants)*: Training continues for an additional 170 epochs on ($\mathbf{z}, \mathbf{x}^1, \mathbf{x}^2$) triplets. The template $\mathbf{z}$ is randomly sampled from a video; $\mathbf{x}^1, \mathbf{x}^2$ are sampled from the same video (up to a 100-frame gap) with strong crop jitter. The backbone and previously trained LoRA modules remain frozen. A new set of extra LoRA modules (rank $r = 64$) and a corresponding prediction head are introduced exclusively for the $\mathbf{x}^2$ stream.

**Inference.** During inference, all LoRATv2 variants leverage frame-wise causal attention with KV caching. The initial template is encoded only once, and its key/value embeddings are cached to prevent re-computation for past frames. Subsequent search regions are cropped around the prior bounding box (area factor 4 for $\mathbf{x}^1$, 5 for $\mathbf{x}^2$).

*Phase 1 ($-224$ variants)*: Tracking is performed using only the $\mathbf{x}^1$ stream.

*Phase 2 ($-378$ variants)*: By default, predictions are derived from the high-resolution $\mathbf{x}^2$ stream. To maintain a temporal context on the past frames, the Key-Value (KV) caches for the $\mathbf{x}^1$ stream are conditionally updated. This update is triggered when a prediction's classification score exceeds a confidence threshold of 0.9. Upon this condition, a new, smaller search region, with dimensions identical to the Phase 1 input, is cropped around the confident prediction and processed to refresh the $\mathbf{x}^1$ stream's cached embeddings.

### 4.2 State-of-the-Art Comparison

We compare our **LoRATv2** with recent Transformer-based trackers on four challenging benchmarks, following their official evaluation protocols. Tab. 1 reports the results.

**LaSOT** [16] is a large-scale benchmark containing 280 long-term test videos. From Tab. 1 and Tab. 2, our *smallest* variant, *LoRATv2-B-224*, achieves a Success (SUC) of 72.0% at 713 *fps*. Meanwhile, *LoRATv2-L-378* sets a new state of the art at 76.1% SUC, outperforming the previous best LoRAT-L$_{378}$ [24] (75.1%).

**TNL2K** [34] is a recently introduced tracking dataset comprising 700 test videos. Our *LoRATv2-L-378* attains 62.4% SUC, improving upon LoRAT-L-378 (62.3%) and confirming the advantage of our temporal modeling capability.

**GOT-10k** [20] consists of 180 test videos and enforces a strict protocol requiring training exclusively on its designated training split. 1 shows that *LoRATv2-L-378* achieves a high AO of 78.2%, approaching ARTrackV2-L (79.5%), thus demonstrating our method's strong generalization when limited to a single dataset.

**VastTrack** [30] is a large-scale benchmark featuring extensive object categories (2,115 classes) to facilitate the development of more general and robust trackers. As shown in Tab. 1, *LoRATv2-L-378*

Table 2: Comparison on efficiency with state-of-the-art Transformer trackers. The values in parentheses for the # params of our trackers represent LoRA and extra components (token type embedding and head), respectively. The speed of all trackers was re-evaluated on our machine.

| Tracker | Speed (*fps*) | MACs (G) | #Params (M) |
|---|---|---|---|
| OSTrack$_{256}$ [43] | 244 | 21.5 | - |
| OSTrack$_{384}$ [43] | 165 | 48.3 | - |
| SeqTrack-B$_{256}$ [7] | 124 | 66 | 89 |
| SeqTrack-L$_{384}$ [7] | 22 | 524 | 309 |
| ROMTrack$_{256}$ [4] | 224 | 35 | 92 |
| ROMTrack$_{384}$ [4] | 141 | 78 | 92 |
| LoRAT-B$_{224}$ [24] | 546 | 30 | 99 (11, 2) |
| LoRAT-B$_{378}$ [24] | 401 | 97 | 99 (11, 2) |
| LoRAT-L$_{224}$ [24] | 255 | 103 | 336 (28, 4) |
| LoRAT-L$_{378}$ [24] | 167 | 325 | 336 (28, 4) |
| LoRATv2-B$_{224}$ | 713 | 25 | 110 (22, 2) |
| LoRATv2-B$_{378}$ | 425 | 81 | 123 (11, 2) |
| LoRATv2-L$_{224}$ | 288 | 85 | 364 (56, 4) |
| LoRATv2-L$_{378}$ | 202 | 268 | 396 (28, 4) |

Table 3: Ablation study on Frame-Wise Causal Attention (FWCA) vs. Fully Self-Attention (FSA), and Stream-Specific LoRA Adapters (SSLA) vs. Shared LoRA. Performance metrics are SUC (Success) and P (Precision) on LaSOT [16] and VastTrack [30]. FPS and MACs denote efficiency.

| Variants | Attention | LoRA Setup | LaSOT | | VastTrack | | FPS | MACs (G) |
|---|---|---|---|---|---|---|---|---|
| | | | SUC (%) | P (%) | SUC (%) | P (%) | | |
| *Based on LoRATv2-B$_{224}$ (Single-Search-Region):* | | | | | | | | |
| ① Baseline | FSA | Shared LoRA | 71.9 | 77.6 | 39.0 | 38.3 | 467 | 49 |
| ② | FSA | SSLA | 72.0 | 77.3 | 38.8 | 38.0 | 452 | 49 |
| ③ | FWCA | Shared LoRA | 71.7 | 77.6 | 38.7 | 38.0 | 713 | 25 |
| ④ Our Model | FWCA | SSLA | **72.0** | **77.9** | **39.1** | **38.7** | **713** | **25** |
| *Based on LoRATv2-B$_{378}$ (Two-Search-Region):* | | | | | | | | |
| ⑤ Baseline | FSA | Shared LoRA | 73.7 | 79.9 | 39.7 | 39.6 | 250 | 137 |
| ⑥ | FSA | SSLA | 73.2 | 78.7 | 39.3 | 39.0 | 237 | 137 |
| ⑦ | FWCA | Shared LoRA | - | - | - | - | 425 | 81 |
| ⑧ Our Model | FWCA | SSLA | **74.3** | **80.9** | **40.6** | **40.8** | **425** | **81** |

achieves $44.2\%$ SUC, surpassing the previous best LoRAT-L$_{378}$ ($43.9\%$) and demonstrating robust performance across diverse categories.

**Efficiency Comparison.** Tab. 2 compares the efficiency of LoRATv2 and leading Transformer trackers. LoRATv2 demonstrates significant improvements in both MACs and practical inference speed (FPS) compared to its predecessor, LoRAT, and other contemporary trackers. For instance, *LoRATv2-B-224* achieves $713\,fps$ with only 25G MACs, outperforming LoRAT-B-224 ($546\,fps$, 30G MACs) in both speed and computational load, while also achieving higher accuracy on LaSOT (Tab. 1 and Tab. 2). This trend holds for larger models as well; *LoRATv2-L-378* runs at $202\,fps$ with 268G MACs, compared to LoRAT-L-378 at $167\,fps$ with 325G MACs. The improved FPS, despite the multi-frame processing and KV cache management, highlights the effectiveness of our architectural optimizations and frame-wise causal attention in reducing redundant computations.

## 4.3 Ablation Studies

We conduct comprehensive ablation studies (Tab. 3) on key components of LoRATv2 using the ViT-Base backbone. All results are reported on LaSOT and VastTrack datasets, along with FPS and MACs. For more ablation experiments, please touch the appendix.

Table 4: Ablation study on the training strategy. We compare our proposed Two-Phase Progressive Training against a standard One-Phase (from-scratch) approach. Performance is evaluated on LaSOT and VastTrack. Training resource consumption (Peak Memory, Time) is also reported.

| Training Strategy | Schedule | LaSOT SUC (%) | VastTrack SUC (%) | Peak Mem. (GB) | Time (h) |
|---|---|---|---|---|---|
| One-Phase | 1x | 69.7 | 36.1 | 17.5 | 16 |
| One-Phase | 2x | 71.3 | 39.0 | 17.5 | 32 |
| **Two-Phase (Ours)** | **1x + 1x** | **74.3** | **40.6** | **15.6** | **19** |

**Frame-Wise Causal Attention (FWCA) and SSLA.** Table 3 analyzes the interplay between the attention mechanism (FWCA vs. standard Fully Self-Attention, FSA) and the LoRA configuration (Stream-Specific vs. Shared). The results reveal a strong synergy between FWCA and SSLA. When paired with SSLA, FWCA delivers substantial efficiency gains without compromising—and often improving—accuracy. For instance, our final single-frame model (④, FWCA+SSLA) nearly doubles the speed of its FSA counterpart (②, 713 vs. 452 FPS) while halving the MACs (25 vs. 49G) and achieving superior performance on VastTrack. This advantage becomes more critical in the multi-frame setting, where the FWCA+SSLA model (⑧) surpasses its FSA equivalent (⑥) in both accuracy (+1.1% SUC on LaSOT) and efficiency.

The necessity of SSLA is most evident when combined with FWCA. While a shared LoRA configuration converges for a single search frame (③), it suffers from training instability and fails to converge in the more complex two-frame setup (⑦). In contrast, SSLA provides the necessary adaptability to manage the informational asymmetry introduced by the causal structure, ensuring robust training and optimal performance. Conversely, with FSA, the benefits of SSLA are less clear and can be slightly detrimental (⑥ vs. ⑤), as the symmetric nature of full attention does not require such stream-specific adaptation.

**Two-Phase Progressive Training.** To validate our curriculum-based training strategy, we compare it against a standard "one-phase" approach where the multi-frame tracker is trained from scratch. As detailed in Table 4, our two-phase method is superior in both effectiveness and efficiency. The model trained progressively achieves a LaSOT SUC of **74.3%**, significantly outperforming the one-phase model even when the latter is trained for twice as long (71.3%). This result highlights the benefit of mastering a simpler task before progressing to a more complex one. Furthermore, our approach is more resource-conscious, reducing training time by 40% (19 vs. 32 hours) and lowering peak GPU memory consumption compared to the extended one-phase baseline. This analysis confirms that progressive training is not merely a heuristic but a more effective and efficient paradigm for developing complex temporal trackers.

## 5 Conclusion

LoRATv2 significantly advances multi-frame object tracking by introducing frame-wise causal attention with KV caching, Stream-Specific LoRA Adapters, and a progressive two-phase training strategy. These innovations lead to state-of-the-art performance with substantially improved computational efficiency and a superior accuracy-FLOPs trade-off, as demonstrated on multiple benchmarks. LoRATv2 offers a powerful and practical solution for real-time tracking with Transformers.

## Acknowledgment

This work was supported in part by the Natural Science Foundation of China (Grant No.62503323). This work was conducted while Liting Lin was a postdoctoral researcher at Pengcheng Laboratory, mentored by Yaowei Wang. Heng Fan participated in this research prior to Yaowei Wang's affiliation with Harbin Institute of Technology and was not supported by any funds for this work.

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
