# OpenReview forum: "LoRATv2: Enabling Low-Cost Temporal Modeling in One-Stream Trackers"
_NeurIPS.cc/2025/Conference — NeurIPS 2025 spotlight_

### Official Review · Reviewer_A63V · 2025-06-27

**Clarity:** 2
**Significance:** 3
**Originality:** 2
**Rating:** 5
**Confidence:** 4

**Summary:**

This paper proposes LoRATv2, a novel framework for efficient multi-frame temporal modeling in Transformer-based trackers. The key contributions are Frame-Wise Causal Attention (FWCA), Stream-Specific LoRA Adapters (SSLA), and two-phase progressive training. Extensive experiments demonstrate state-of-the-art accuracy–FLOPs trade-offs across multiple tracking benchmarks.

**Questions:**

1.What level of performance can be achieved using only Phase 1 in low-compute scenarios?
2.Does caching multiple frames in the KV cache lead to GPU memory or efficiency bottlenecks, or cause performance degradation?
3.How do different LoRA rank values impact performance?

**Ethical Concerns:**

["NO or VERY MINOR ethics concerns only"]

**Final Justification:**

The author response has addressed most of my concerns. So I tend to raise the final rating.

**Limitations:**

1.Due to the causal attention mechanism, reliance on historical frames in occlusion and fast-motion scenarios can lead to tracking errors and degrade tracking accuracy.
2.As the number of historical frames increases, the KV cache size also grows, which may cause GPU memory pressure in long-sequence scenarios.

**Quality:**

3

**Strengths And Weaknesses:**

Strengths:
1.Integration of causal attention into one-stream Transformer trackers with KV caching, greatly reducing temporal modeling cost.
2.SSLA provides stream-specific adaptation without unfreezing the ViT backbone, incurring zero extra inference cost.
3.The two-phase progressive curriculum reduces memory/time overhead in multi-frame training while improving long-term tracking.

Weaknesses:
1.The choice of LoRA rank is only reported in Section 4.1 without any ablation study to justify the selected rank size.
2.The discussion of the KV cache is insufficient—for example, what is the maximum cache capacity and how much memory does it consume?
3.The Phase 1 models ablation experiments (1 and 3) do not adequately demonstrate the effectiveness of the FWCA and SSLA modules. Could you please supplement with additional experiments involving self-attention + SSLA?

---

> ### Author Rebuttal · Authors · 2025-07-31
>
> Thank you for your valuable feedback and insightful questions. We provide our responses below.
>
> ---
> **Q1: [Ablation on LoRA Rank]** The choice of LoRA rank is reported without an ablation study to justify the selected rank size. How do different LoRA rank values impact performance?
>
> **A1:** Thanks for the comment. Our choice of LoRA rank $r=64$ was inherited from the hyperparameters used in the original LoRAT.
>
> To verify if this is the suitable choice for the LoRATv2 framework, we conducted an ablation study on both our phase-1 (**LoRATv2-B-224**) and phase-2 (**LoRATv2-B-378**) models with various LoRA ranks. The results are presented below:
>
> | Model | LoRA Rank ($r$) | LaSOT SUC / P (%) | TNL2k SUC / P (%) | Trainable Parameters |
> | :------------ | :---:  |       :---:     |      :---:      | :---: |
> | LoRATv2-B-224 |  16    |   70.1 / 74.8   |   56.4 / 57.5   |  ~7.7M  |
> |               |  32    |   70.6 / 75.5   |   56.6 / 57.8   |  ~13.0M |
> |               | **64** | **72.0 / 77.9** | **59.6 / 62.7** | **~23.6M** |
> |               |  128   |   71.8 / 77.6   |   59.4 / 62.6   |  ~44.8M |
> | LoRATv2-B-378 |  16    |   73.0 / 79.0   |   60.1 / 63.3   |  ~5.0M |
> |               |  32    |   73.1 / 79.6   |   59.9 / 63.2   |  ~7.7M |
> |               | **64** | **74.3 / 80.9** | **60.9 / 64.9** | **~13.0M** |
> |               |  128   |   73.7 / 80.3   |   59.9 / 63.5    |  ~23.6M |
>
> The results show that $r=64$ offers the best trade-off between accuracy and parameter efficiency for our model.
>
> We will include the above analysis and results in revision. Again, thanks!
>
> ---
> **Q2: [Details on KV Cache]** The discussion of the KV cache is insufficient. What is the maximum cache capacity, how much memory does it consume, and does caching multiple frames lead to bottlenecks or performance degradation?
>
> **A2:**
> Thanks and we agree that a thorough explanation of the KV cache's practical impact is essential.
>
> *   **Cache Capacity and Memory Consumption:** The memory footprint of the KV cache is modest. For instance, caching the template and one past search frame in our **LoRATv2-B-378** model consumes only **54.4MB**, and for the larger **LoRATv2-L-378** model, it is **145.6MB**. This constitutes a small and manageable fraction of total GPU memory, which is negligible for most tracking scenarios. The cache capacity is not a hard limit and can be extended to retain more historical frames.
>
> *   **Bottlenecks and Performance Degradation:** Caching multiple frames does not lead to significant bottlenecks or performance degradation. This is for two primary reasons:
>     1.  **Optimized Kernels:** We leverage modern fused attention kernels (e.g., FlashAttention), which are specifically designed to handle such memory access patterns efficiently. These kernels effectively overlap memory operations (i.e., fetching the KV cache from memory) with computation, thereby hiding most of the memory access latency.
>     2.  **Net Computational Gain:** The KV cache provides a substantial net computational gain. Because attention is causal, embeddings for past frames are immutable—they are not influenced by any subsequent frames. They can be computed once, cached, and reused without ever needing to be re-encoded. This massive saving in computation far outweighs the minor cost of memory access.
>
> *   **Performance with More Frames:** To empirically demonstrate that our framework scales gracefully, we progressively extended our LoRATv2-B model to incorporate up to three past search frames. As shown in the table below, adding more frames to the cache consistently **improves** tracking accuracy while only minimally impacting efficiency, confirming that caching does not cause performance degradation.
>
> | # Cached Past Frames (Template + Search) | LaSOT SUC / P (%) | VastTrack SUC / P (%) | MACs (G) | FPS |
> | :--- | :---: | :---: | :---: | :---: |
> | 1 + 1 (LoRATv2-B-378) | 74.3 / 80.9 | 40.6 / 40.8 | 81 | 172 |
> | 1 + 2 | 74.6 / 81.2 | 41.8 / 42.7 | 84 | 170 |
> | 1 + 3 | 75.0 / 81.5 | 42.1 / 42.9 | 88 | 169 |
>
> These results confirm that not only does caching more frames avoid bottlenecks, it also actively improves performance with a very graceful degradation in FPS. We will provide these details about the KV cache by including the above analysis in revision. Again, thanks!
>
> ---
> **Q3: [More Ablations for FWCA and SSLA]** The Phase 1 ablation experiments do not adequately demonstrate the effectiveness of FWCA and SSLA. Could you please supplement with an experiment involving standard self-attention + SSLA?
>
> **A3:**
> Thanks for this thoughtful comment. This experiment was conducted and included in our appendix but omitted from the main text due to space limitations. To provide a comprehensive view, we present the full ablation table below and will ensure the designs for Phase 1 and 2 in Table 3 are consistent in the revision.
>
> | Model | Attention | LoRA Setup | LaSOT SUC / P (%) | VastTrack SUC / P (%) | FPS | MACs (G) |
> | :--- | :--- | :---:| :---: | :---: | :---: | :---: |
> | LoRATv2-B-224 | Full Self-Attn | Shared LoRA | 71.9 / 77.6 | 39.0 / 38.3 | 163 | 49 |
> |  | Full Self-Attn | SSLA | 72.0 / 77.3 | 38.8 / 38.0 | 158 | 49 |
> |  | **FWCA** | **SSLA** | **72.0 / 77.9** | **39.1 / 38.7** | **249** | **25** |
> |
> | LoRATv2-B-378 | Full Self-Attn | Shared LoRA | 73.7 / 79.9 | 39.7 / 39.6 | 101 | 137 |
> |  | Full Self-Attn | SSLA | 73.2 / 78.7 | 39.3 / 39.0 | 96 | 137 |
> |  | **FWCA**  | **SSLA** | **74.3 / 80.9** | **40.6 / 40.8** | **172** | **81** |
>
> With standard Full Self-Attention, the bidirectional self-attention are symmetric, so moving from a Shared LoRA to SSLA provides only a marginal benefit. However, the combination of **FWCA and SSLA** not only yields comparable or better accuracy but also dramatically improves efficiency (e.g., 249 vs. 163 FPS for the -224 model). For the two-search-frame model, the benefits are even clearer, with our final design (FWCA+SSLA) substantially outperforming all other variants in both accuracy and efficiency.
>
> We will include more detailed analysis in revision. Again, thanks!
>
> ---
> **Q4: [Performance of Phase 1 Model]**
> What level of performance can be achieved using only the Phase 1 model in low-compute scenarios?
>
> **A4:**
> Thanks for this insightful comment. As suggested, to further demonstrate its effectiveness in low-compute scenarios, we conducted an ablation of our method for CPU inference. Specifically we trained a **LoRATv2-S-224** variant by accordingly adopting a smaller ViT-Small backbone. For a fair and direct comparison against leading lightweight trackers, all models were re-evaluated on the same consumer-grade CPU (Intel Ultra 7 265k).
>
> | Model Name | LaSOT SUC / P (%) | VastTrack SUC / P (%) | UAV123 SUC / P (%) | MACs (G) | CPU FPS |
> |:---:|:---:|:---:|:---:|:---:|:---:|
> | HiT-Small [1]          | 60.5 / 61.5 | - / -| 63.3 / -  | 1.13 | 60.3 |
> | MixFormerV2-S [2] | 60.6 / 60.4 | -/ - | 65.8 / 86.8 | 4.4 | 61.2 |
> | LoRATv2-S-224 | 69.9 / 75.3 | 36.9 / 35.2 | 71.5 / 91.4 | 6.9 | 40 |
>
> The results show that our architecture also can be served as a highly competitive option within the domain of efficient visual tracking in low-compute scenarios. We will include the analysis in revision. Again, thanks!
>
> [1] Kang, B., Chen, X., Wang, D., Peng, H., & Lu, H. (2023). Exploring Lightweight Hierarchical Vision Transformers for Efficient Visual Tracking. In: ICCV.
>
> [2] Cui, Y., Song, T., Wu, G., & Wang, L. (2023). MixFormerV2: Efficient Fully Transformer Tracking. In: NeurIPS.

---

> > ### Comment · Reviewer_A63V · 2025-08-06
> >
> > Thanks for the response, which have addressed most of my concerns. I tend to raise the final rating.

---

> > > ### Author Response · Authors · 2025-08-07
> > >
> > > We thank the reviewer for their positive reassessment. We are grateful for their time and constructive feedback.

---

### Official Review · Reviewer_DMUb · 2025-07-01

**Clarity:** 4
**Significance:** 4
**Originality:** 3
**Rating:** 5
**Confidence:** 3

**Summary:**

The paper proposes LoRATv2, an efficient multi-frame Transformer tracker that combines three ideas: (i) frame-wise causal attention plus KV-caching to cut the quadratic cost of vanilla self-attention, (ii) stream-specific LoRA adapters that let template and search streams specialize while keeping the ViT backbone frozen, and (iii) a two-phase curriculum that first trains on single-frame inputs and then extends to multi-frame ones.

**Questions:**

See Weaknesses.

**Ethical Concerns:**

["NO or VERY MINOR ethics concerns only"]

**Final Justification:**

The author response has addressed most of my concerns; therefore, I will maintain my original rating.

**Limitations:**

yes

**Paper Formatting Concerns:**

I did not notice any major deviations from the NeurIPS 2025 formatting guidelines.

**Quality:**

3

**Strengths And Weaknesses:**

Strengths
1. Well-targeted efficiency innovation. The frame-wise causal mask plus KV-cache gives a clean O(N) path length without redesigning the backbone.
2. Stream-specific LoRA keeps training cost low and lets the authors reuse large off-the-shelf ViTs.
3. The paper is well-written and easy to follow, with clear visualizations.

Weaknesses
1. Table 1 needs fresher baselines.
2. OSTrack speed anomaly in Table 2. Although OSTrack shows lower MACs than LoRATv2, its FPS is lower.
3. Minor English issues. “state-of-the-art” appears both hyphenated and as “state of the art”; consider unifying.

---

> ### Author Rebuttal · Authors · 2025-07-31
>
> We appreciate your positive assessment and helpful suggestions. We address your points below.
>
> **Q1: [Fresher Baselines in Table 1]** Table 1 needs fresher baselines.
>
> **A1:**
> Thank you for the suggestion. We will update main comparison table (Table 1) in the revision by including more recent state-of-the-art trackers, as shown below:
>
> | Model | Source |LaSOT SUC / P(%) | TNL2k SUC / P(%) | MACs (G) |
> |:------------ | :---:  |       :---:     |      :---:      | :---: |
> | LMTrack$_{384}$ [1] | AAAI 2025 | 73.2 / 81.0 | - / - | 69 |
> | MambaLCT-384 [2] | AAAI 2025 | 73.6 / 81.6 | 58.5 / - | 58 |
> | LoRATv2-B-378 | Ours | 74.3 / 80.9 | 60.9 / 64.9  | 81 |
> ||
> | ARPTrack-L$_{384}$  [3] | CVPR 2025 | 74.2 / 81.7 |  -  | - |
> | SeqTrack-L$_{384}$  + MPT [4] | ICML 2025 | 73.9 / 81.0 | 60.4 / 64.2 | - |
> | LoRATv2-L-378 | Ours | 76.1 / 83.1 | 62.4 / 67.7 | 268 |
>
> [1] Xu, C., Zhong, B., Liang, Q., Zheng, Y., Li, G., & Song, S. (2025). Less is More: Token Context-aware Learning for Object Tracking. In: AAAI.
>
> [2] Li, X., Zhong, B., Liang, Q., Li, G., Mo, Z., & Song, S. (2025). MambaLCT: Boosting Tracking via Long-term Context State Space Model. In: AAAI.
>
> [3] Liang, S., Bai, Y., Gong, Y., & Wei, X. (2025). Autoregressive Sequential Pretraining for Visual Tracking. In: CVPR.
>
> [4] Zhao, J., Chen, X., Yuan, Y., Felsberg, M., Wang, D., & Lu, H. (2025). Efficient Motion Prompt Learning for Robust Visual Tracking. In: ICML.
>
> ---
> **Q2: [OSTrack Speed Anomaly in Table 2]** Although OSTrack shows lower MACs than LoRATv2, its FPS is lower. Explain this anomaly.
>
> **A2:**
> Thanks for this careful comment. The discrepancy between theoretical MACs and practical FPS arises primarily from architectural differences affecting hardware utilization. OSTrack's "Early Candidate Elimination" strategy, while reducing MACs, introduces sequential logic and conditional branching that are less amenable to the massive parallelization of modern GPUs, thus increasing latency. And in our testing, it requires higher-precision (float32) arithmetic for stable performance, which is inherently slower. Furthermore, its convolutional head introduce higher FLOPs compared with our MLP-based head.
>
> LoRATv2's design is highly parallelizable and optimized for fused GPU kernels, leading to higher throughput (FPS) despite the slightly higher MAC count.
>
> We will include the above clarification in revision. Thanks again!
>
> ---
> **Q3: [Minor English Issues]** The term “state-of-the-art” appears in different forms.
>
> **A3:** Thank you for pointing this out. We will carefully proofread the entire manuscript to ensure consistent terminology in the whole manuscript.

---

> > ### Comment · Reviewer_DMUb · 2025-08-06
> >
> > I would like to thank the authors for the rebuttal. Most of my concerns are addressed. Currently, I would like to keep my original score.

---

> > > ### Author Response · Authors · 2025-08-06
> > >
> > > We sincerely thank the reviewer for taking the time to review our manuscript and providing constructive feedback to improve our manuscript.

---

### Official Review · Reviewer_ZLNb · 2025-07-02

**Clarity:** 3
**Significance:** 3
**Originality:** 2
**Rating:** 4
**Confidence:** 5

**Summary:**

First, this paper addresses the computational complexity of standard attention operations in LoRAT v1. Specifically, when processing high-resolution or multi-frame inputs, the standard attention mechanism incurs substantial computational overhead during inference. To resolve this, the paper reframe tracking as an autoregressive sequence prediction task and propose a frame-wise causal attention mechanism, where each frame exclusively attends to preceding frames. This approach is further accelerated through key/value caching, significantly reducing inference latency.

Second, while causal attention mitigates computational costs, it introduces feature inconsistency across frames due to varying receptive fields. The paper counteract this by designing Stream-Specific LoRA Adapters, which allocate dedicated LoRA modules to distinct input frames. This ensures parameter-efficient feature alignment while maintaining temporal coherence.

Third, to scale the model for multi-frame inputs, a two-phase progressive training strategy (inspired by curriculum learning) is introduced. Compared to direct end-to-end training on multi-frame data, this phased approach reduces training complexity.

**Questions:**

1. The Model Variants section in 4.1 is unclear. The model's input settings for different phases and resolutions should be clearly specified.

2. Table 2 shows the parameters of different models in terms of efficiency. The parameters of LoRAT are consistent across different resolutions, but those of LoRAT v2 are not. Could you please explain the reason for this?

3. Could you please explain the reasons for the failure of ② in Table 3 to converge as much as possible?

**Ethical Concerns:**

["NO or VERY MINOR ethics concerns only"]

**Final Justification:**

The authors have addressed most of my concerns. Regarding the originality of the FWCA method, although the authors have further elaborated on the differences between FWCA and S-MAM, I personally find the distinction not sufficiently clear. However, considering its differentiated application in AR tasks and the inclusion of comparative experiments in the manuscript, I retain partial reservations. Therefore, I will maintain my original score.

**Limitations:**

yes

**Paper Formatting Concerns:**

/

**Quality:**

3

**Strengths And Weaknesses:**

1. Quality: The paper presents technically sound methodologies with substantial experimental validation. However, additional experiments are required to strengthen the conclusions:

    • Comparative analysis of direct multi-frame training: Quantitative results (accuracy/FPS) and resource consumption (GPU hours/memory) versus the proposed two-stage strategy, demonstrating the necessity of phased training.

    • Scalability validation of SSLA: Performance evaluation with more frame inputs to substantiate Stream-Specific LoRA Adapters' efficacy in large-scale scenarios.

2. Clarity While generally well-structured, the manuscript requires little revisions:

- The type emb in Figure 2 comes from the LoRAT v1 paper, but it is not mentioned in the revisiting section of this paper, so it needs to be briefly explained again.


- Table 3 conducts ablation experiments on model design. First, as a personal suggestion, if the final model design is used as the baseline for the experiment, the Variants column should use “-” to indicate changes in model design instead of “+”. For example, “+ w/o SSLA” should be “- SSLA”. Second, the ablation designs for Variants in phases 1 and 2 should be as consistent as possible.


3. Significance: The work significantly addresses the computational bottleneck of ViT-based trackers during inference, particularly for high-resolution/multi-frame use cases.

4. Originality: The core contributions are primarily engineering-driven enhancements rather than theoretical innovations:

- Frame-wise causal attention: While effective, this mirrors prior autoregressive designs and shares operational similarities with S-MAM in MixFormer.

- Key-value caching: Positioned as an implementation optimization (also seen in MixFormer's public codebase) rather than a novel methodological contribution.

---

> ### Author Rebuttal · Authors · 2025-07-31
>
> Thank you for your detailed and constructive feedback. We address your questions and concerns below.
>
> **Q1: [Comparative analysis of training strategies]**
> Provide a comparative analysis of direct multi-frame training versus the proposed two-stage strategy, covering quantitative results (accuracy/FPS) and resource consumption (GPU hours/memory) to demonstrate the necessity of phased training.
>
> **A1:**
> We appreciate the reviewer's valuable suggestion. To highlight the necessity and advantages of our progressive training strategy, we compare it with a standard "from-scratch" multi-frame training baseline. All methods are based on LoRATv2-B-378, the only difference lies in the training strategy.
>
> |Training Strategy|Schedule|LaSOT SUC (%)|VastTrack SUC (%)|Inference FPS|Peek Memory (GB)|Time (h)|
> | :- | :-| :-:| :-: |:-:|:-:|:-:|
> |One-Phase|1x|69.7|36.1|172|17.5|16|
> |One-Phase|2x|71.3|39.0|172|17.5|32|
> |**Two-Phase (Ours)**|**1x + 1x**|**74.3**|**40.6**|**172**|**15.6**|**19**|
>
> In this experiment:
> *   **"One-Phase (1x)"**: Direct two-search-frame training for 170 epochs.
> *   **"One-Phase (2x)"**: Doubles to 340 epochs.
> *   **"Two-Phase (1x+1x)"** is our proposed method: 170 epochs for the single-search-frame model, followed by an additional 170 epochs for the two-search-frame model.
>
> The results clearly validate the advantages of our approach:
>
> 1.  **Higher Accuracy:** Our two-phase strategy achieves **74.3% SUC** on LaSOT, significantly outperforming the from-scratch model (71.3% SUC for "One-Phase (2x)"). This highlights the effectiveness of the curriculum-like learning process, which guides the model evolving from a simpler to a more complex task.
>
> 2.  **Greater Efficiency:** Compared to the "One-Phase (2x)" baseline, our method requires substantially less training time (**19 hours vs. 32 hours**) and consumes less peak GPU memory (**15.6 GB vs. 17.5 GB**), primarily because fewer parameters are trainable in the second phase.
>
> We will include this comparative analysis in the revision. Again, thanks!
>
> ---
> **Q2: [Scalability of SSLA]**
> Provide a performance evaluation with more frame inputs to substantiate the scalability and efficacy of Stream-Specific LoRA Adapters in large-scale scenarios.
>
> **A2:**
> We appreciate this insightful question regarding the scalability of our framework.
>
> To empirically validate the scalability, we progressively extended our LoRATv2-B-378 model to incorporate up to three past search frames. Each new frame was managed by a new, progressively trained SSLA module.
>
> The results are presented below:
>
> |# Cached Past Frames (Template + Search)|LaSOT SUC / P (%)|VastTrack SUC / P (%)|MACs (G)|FPS|
> |:-|:-:|:-:|:-:|:-:|
> |1 + 1 (LoRATv2-B-378)|74.3 / 80.9|40.6 / 40.8|81|172|
> |1 + 2|74.6 / 81.2|41.8 / 42.7|84|170|
> |1 + 3|75.0 / 81.5|42.1 / 42.9|88|169|
>
> This experiment reveals two key advantages of our approach:
>
> 1.  **Consistent Performance Gains:** Tracking accuracy improves consistently as more frames are added. This demonstrates that our model, guided by the specialized adaptations from SSLA, can effectively leverage longer temporal history for more robust predictions, with particularly notable gains on the challenging VastTrack dataset.
>
> 2.  **Computational Efficiency:** The computational cost grows very slowly. Scaling from one to three past search frames increases MACs by only ~9% (from 81G to 88G), with a minimal impact on FPS. This high efficiency stems from our architecture: the most computationally intensive ViT layers (Attention and MLP linear projections) process only the newest search frame's tokens. The complexity of the frame-wise causal attention mechanism grows linearly with cached frames, and its contribution to total MACs is minor.
>
> We will include the above analysis and results in reivions. Thanks again!
>
> ---
> **Q3: [Clarity of “type emb.”]**
> The "type emb" in Figure 2 is from LoRAT v1 but is not mentioned in the revisiting section.
>
> **A3:** Thanks for this careful comment. We will add a clear explanation of this component in the revision.
>
> ---
> **Q4: [Clarity of Table 3]** Table 3 conducts ablation experiments on model design. First, as a personal suggestion, if the final model design is used as the baseline for the experiment, the Variants column should use “-” to indicate changes in model design instead of “+”. For example, “+ w/o SSLA” should be “- SSLA”. Second, the ablation designs for Variants in phases 1 and 2 should be as consistent as possible.
>
> **A4:**
> Thanks for suggestions. We'll update Table 3 to "-" notation and ensure Phase 1/2 consistency.
>
> ---
> **Q5: [Originality of FWCA and KV Cache]**
> The core contributions are primarily engineering-driven enhancements rather than theoretical innovations: Frame-wise causal attention: While effective, this mirrors prior autoregressive designs and shares operational similarities with S-MAM in MixFormer. Key-value caching: Positioned as an implementation optimization (also seen in MixFormer's public codebase) rather than a novel methodological contribution.
>
> **A5:**
> Thanks for this insightful comment.
>
> Our Frame-Wise Causal Attention (FWCA) builds on autoregressive principles to enable efficient temporal modeling without recomputing historical frames.
> However, autoregressive models are prone to error accumulation[1]: early prediction mistakes propagate and amplify over time, especially during early training phases.
> To specifically mitigate this issue, we introduced novel Stream-Specific LoRA Adapters (SSLA) and the two-phase progressive training strategy, which together ensure stable convergence and robust temporal modeling.
>
> This design sets our approach apart from methods like S-MAM in MixFormer. While S-MAM prunes target-to-search cross-attention and attention among target templates, effectively treating multiple templates as a reference bank for localization, our Frame-Wise Causal Attention establishes a genuine autoregressive history. By sequentially processing frames, it captures the target's evolving appearance and state, which strengthens tracking robustness and paves the way for future advancements in motions-aware or occlusions-aware trackers. As evidenced in our experiments, this design leads to superior performance compared to S-MAM.
>
> Although KV caching serves as an known optimization to accelerate autoregressive inference, akin to implementations in other codebases, our key innovation lies in integrating it with SSLA and progressive training to address error accumulation.
>
> Again, we thank the reviewer for these helpful comments. We will include the above explanations and more discussions in revision to clarify the difference of our method with other methods. Thanks!
>
> [1] Yin, T., Zhang, Q., Zhang, R., Freeman, W. T., Durand, F., Shechtman, E., & Huang, X. (2025). From Slow Bidirectional to Fast Autoregressive Video Diffusion Models. In CVPR.
>
> ---
> **Q6: [Clarity of Model Variants]** The "Model Variants" section (4.1) is unclear. The model's input settings for different phases and resolutions should be clearly specified.
>
> **A6:**
> We apologize for the lack of clarity. We will revise it as follows:
> *   **LoRATv2-B/L-224 (Phase 1 Models):**
>     *   Backbone: ViT-Base/ViT-Large
>     *   Template ($\mathbf{z}$): $224 \times 224$
>     *   Search Region ($\mathbf{x}^1$): $224 \times 224$
> *   **LoRATv2-B/L-378 (Phase 2 Models):**
>     *   Backbone: ViT-Base/ViT-Large
>     *   Template ($\mathbf{z}$): $224 \times 224$
>     *   Past Search Region ($\mathbf{x}^1$): $224 \times 224$
>     *   Current Search Region ($\mathbf{x}^2$): $378 \times 378$
>
> ---
> **Q7: [Parameters in Efficiency Table]**
> In Table 2, the parameters of LoRAT are consistent across different resolutions, but those of LoRATv2 are not. Explain the reason for this.
>
> **A7:**
> Thanks for this careful comment. The difference stems from a core design choice:
> *   **LoRAT** uses a *single* set of LoRA adapters whose weights are shared between the template and search streams. Thus, changing input resolution does not alter the number of trainable parameters.
> *   **LoRATv2**, with our proposed SSLA, uses *dedicated* LoRA adapters for each stream. The "-378" models introduce an additional stream for the second search frame ($\mathbf{x}^2$), which requires its own new LoRA adapter and a new prediction head. This is the source of the increased parameter count for our larger, multi-search-frame models.
>
> We will add a clear explanation of this point in Section 4.2 when discussing the efficiency comparison. Again, thanks!
>
> ---
> **Q8: [Convergence Failure]** Explain the reasons for the failure of variant ② in Table 3 (FWCA with Shared LoRA) to converge.
>
> **A8**: The convergence failure is caused by the functional asymmetry introduced by FWCA, which cannot adequately model with a simple shared LoRA. If a shared LoRA is tasked with learning two functionally distinct adaptations using a single set of weights, it woule lead to unstable optimization. Our SSLA adapt such asymmetry in network structure.
>
> To validate this, we conducted a detailed ablation study:
>
> |Attention|Training Method|Converged?|LaSOT SUC / P (%) |VastTrack SUC / P (%)|
> |:-:|:-:|:-:|:-:|:-:|
> |Self-Attn | Full Finetune | Yes | 71.2 / 76.9 | 39.0 / 38.4 |
> | Self-Attn | Shared LoRA | Yes | 71.9 / 77.6 | 39.0 / 38.3 |
> |Self-Attn|SSLA|Yes|72.0 / 77.3|38.8 / 38.0|
> |FWCA|Full Finetune|Yes|69.4 / 73.9|36.0 / 34.0|
> |FWCA|Shared LoRA|**No**|-|-|
> |**FWCA**|**SSLA**|**Yes**|**72.0 / 77.9**|**39.1 / 38.7**|
>
> The results are revealing:
> *   With symmetric Full Self-Attention, all methods converge, as Shared LoRA is sufficient.
> *   With asymmetric FWCA, Shared LoRA fails entirely. Full fine-tuning can adapt with the asymmetry but results in suboptimal performance.
> *   Only the combination of **FWCA and SSLA** both converges reliably and achieves the best performance, demonstrating that SSLA is an essential component of our framework.
>
> We will include the above clarification and results in revision. Again, thanks for this comment!

---

> > ### Comment · Reviewer_ZLNb · 2025-08-06
> >
> > Thanks for the authors' rebuttal, which have addressed most of previous concerns. So I would like to keep my original score.

---

> > > ### Author Response · Authors · 2025-08-06
> > >
> > > We thank the reviewer again for their time and valuable feedback.

---

### Decision · Program_Chairs · 2025-09-17

**Decision:**

Accept (spotlight)

**Comment:**

This paper addresses the computational complexity of standard attention in LoRAT v1, which becomes costly for high-resolution or multi-frame inputs. To mitigate this, tracking is reformulated as an autoregressive sequence prediction task, and a frame-wise causal attention mechanism is introduced, where each frame attends only to preceding ones. With key/value caching, inference latency is further reduced. Since causal attention can cause feature inconsistencies across frames, the authors propose Stream-Specific LoRA Adapters, assigning dedicated LoRA modules to each frame for efficient feature alignment and temporal coherence. Finally, a two-phase progressive training strategy inspired by curriculum learning is presented to scale the model to multi-frame inputs, reducing training complexity compared to direct end-to-end training.
The paper has received two accepts and one borderline accept. All reviewers noted that the method is novel and the experimental evaluation is sufficiently strong to support acceptance. Therefore, I recommend accepting this paper.